# Pose-Aware Proxies for Unsupervised Marine Wildlife Re-Identification

## Abstract

Scaling wildlife re-identification remains challenging due to reliance on expert photo-ID and large labeled datasets. In Malapascua, Philippines, divers capture abundant unlabeled footage of endangered thresher sharks, motivating an unsupervised solution. We curate a structured dataset of thresher-shark dive videos organized by co-occurrence and track-based local identities, and introduce pose-aware proxies, which arecoarse orientation labels that provide weak viewpoint supervision within a clustering-based contrastive framework. We evaluate without global identity labels using three field-aligned metrics: within-track consistency (WTC), co-occurrence recall (CoR@k), and mutual-exclusion error (MEError@k). On our dataset, the TP6 variant (excluding ambiguous "Others") improves temporal stability (-23.5% WTC vs Base) and reduces impostor matches (MEError down 19.4% @1, 46.0% @5, 33.7% @10), while slightly lowering CoR at small k (gap narrows by k=10). These results show that pose-conditioned guidance extends proxy-based unsupervised learning to unconstrained ecological video, prioritizing precision over immediate recall, and they isolate cross-pose matching as a key open challenge for future work.

## 1 Introduction

Visual re-identification (re-ID) links repeated observations of the same individual and underpins abundance estimates in marine-wildlife monitoring, yet the task is hampered by scarce labels and rapidly changing viewpoints in opportunistic video. Recent unsupervised pipelines therefore rely on clustering-contrastive loops whose supervision is provided by proxies which are group-level anchors summarizing subsets of embeddings. Whereas camera-aware proxies condition these anchors on fixed camera IDs, we instead exploit pose-aware proxies that partition embeddings by coarse orientation (e.g., left, right, front-left), treating viewpoint as a structured nuisance and mirroring the camera-conditioning strategy used in O2CAP and related methods (Wang et al., 2021; 2022; Li et al., 2022). This shift is motivated by two mismatches between ecological footage and standard assumptions: (i) handheld, drifting cameras preclude reliable camera IDs, and (ii) global identity labels are typically absent, making classical CMC/mAP metrics inapplicable.

Consequently, the field lacks a simple mechanism to substitute viewpoint conditioning for camera IDs and an evaluation protocol that reflects within-dive structure without global IDs. We address these gaps by operationalizing coarse pose labels as drop-in supervision for Transformer-based multi-granular frameworks, and by proposing three weakly supervised metrics: Within-Track Consistency (temporal stability), Mutual-Exclusion Error@k (same-dive impostor suppression), and Co-Occurrence Recall@k (cross-subclip linkage) that collectively capture precision-recall trade-offs in this setting. Using a curated dataset of identity-pure thresher-shark subclips from Kimud Shoal, we show that excluding ambiguous "Others" frames yields the best balance of stability and precision, thereby clarifying when and how pose granularity matters.

Our contributions are as follows:

- (i) pose-aware proxies compatible with TMGF pipelines,
- (ii) an ecological evaluation suite aligned to dive-local structure,
- (iii) a realistic underwater case study, and

- (iv) a systematic analysis of pose granularity that informs future unsupervised wildlife re-ID designs.

## 2 RELATED WORK

Early animal re-identification (Re-ID) systems largely relied on manually engineered, pattern-matching pipelines that exploited individually distinctive markings visible in photographs. Foundational studies on cheetahs, whale sharks, and gray seals, for example, matched new observations to known individuals by comparing spot, pigmentation, and pelage pattern cues (Kelly, 2001; Arzoumanian et al., 2005; Karlsson et al., 2005). The core premise was that such textures are sufficiently stable over time to function as natural "biometric" signatures. Building on that template, similar methods were extended to other animals with including stripe-patterned and spotted species such as tigers and spotted raggedtooth sharks (Hiby et al., 2009; Van Tienhoven et al., 2007). These pipelines depend on the long-term stability and visibility of the markings and are sensitive to factors such as pose changes, partial occlusion, illumination, and life-stage or seasonal appearance shifts. They also often require nontrivial human effort for annotation and verification. Consequently, the absence of prior studies or distinctive markings in other species renders handcrafted photo-ID labor-intensive and limited in scope. This motivated the shift toward supervised learning approaches that extend animal re-ID beyond species-specific pipelines to more generalizable solutions. Early work on terrestrial animals, such as elephants, leveraged curated flank images and anatomical landmarks but was limited by pose variability and background clutter (Körschens et al., 2018), later improved through part-based alignment strategies (Yu et al., 2024). In marine settings, researchers successfully trained models on contour and shape-based methods, including manta rays (Moskvyak et al., 2019), dolphins (Thompson et al., 2019), and great white sharks (Hughes & Burghardt, 2017). More recently, species-agnostic frameworks have emerged, fueled by large community-contributed datasets and transferable feature learning methods. These models, such as MegaDescriptor, ALFRE-ID, and MiewID, demonstrated strong cross-species generalization by learning robust local and global representations without heavy reliance on species-specific heuristics (Čermák et al., 2023; Nepovinnykh et al., 2024; Otarashvili et al., 2024). However, across all settings, supervised learning remains constrained by its dependency on costly identity-labeled datasets and susceptibility to domain shifts, motivating the shift toward self-supervised approaches that scale without manual annotation.

Unsupervised re-ID is crucial for animal studies, as manual labeling is prohibitively costly, expert annotators are scarce in regions where endangered species live, and conservation programs often face severe funding constraints. In Malapascua, for example, recreational divers and dive shops generate abundant video footage of thresher sharks, yet aggregating and labeling this material is difficult, especially given the absence of established photo-ID markers for the species. Leveraging such unlabeled imagery, unsupervised methods offer a scalable solution for ecological monitoring and conservation. While unsupervised animal re-ID has seen limited progress, unsupervised human re-ID has advanced rapidly, developing clustering and contrastive-based frameworks that progressively refine identity representations without ground-truth labels. Clustering-based unsupervised re-identification has progressively shifted from naive pseudo-labeling to more structured forms of supervision. Early methods treated clusters as entire identities, but this collapsed under large intra-class variance caused by viewpoint changes. Camera-Aware Proxies (CAP) addressed this by conditioning clusters on camera IDs, stabilizing assignments through intra-camera contrastive losses and balanced sampling (Wang et al., 2021). O2CAP extended this principle by dynamically refreshing proxies with an offline-online association scheme, discarding redundant intra-camera losses while generating stronger positives and harder negatives (Wang et al., 2022). Together, these studies established the importance of structured, view-conditioned proxies for suppressing noise in contrastive objectives. In parallel, several approaches emphasized the refinement of pseudo labels at finer granularity. ICE pursued compactness by contrasting anchors with their hardest positives while injecting soft pairwise-similarity labels to remain robust to augmentation noise (Chen et al., 2021). PPLR further showed that cross-checks between global and part features, via Part-Guided Label Refinement and Agreement-Aware Label Smoothing, were critical when pose variation was high (Cho et al., 2022). At a larger scale, Cluster Contrast moved beyond instance-level dictionaries, aligning contrastive loss with cluster centroids and enforcing temporal coherence through momentum updates (Dai et al., 2023). These directions converge on the insight that robust unsupervised re-ID requires both stable pseudo labels and mechanisms to mitigate variance across space and time.

More recently, backbone design has proven equally important. TMGF integrated O2CAP-style proxy losses with a multi-branch Vision Transformer, learning global tokens alongside uniformly striped part tokens (Li et al., 2022). This multi-granular representation significantly narrowed the gap to supervised methods, highlighting the synergy between richer feature hierarchies and structured contrastive learning. However, across these threads, key assumptions remain: proxies rely on fixed camera viewpoints, pseudo-label refinements presuppose roughly aligned poses, and memory-based methods assume stable scene statistics. In ecological video, where cameras drift, animal poses vary wildly, and visibility conditions fluctuate, these premises fail. To bridge this gap, we propose pose-aware proxies that replace camera IDs with coarse orientation labels, enabling structured supervision under unconstrained viewpoints. Built atop O2CAP and enhanced with multi-granular transformer features, our method preserves the benefits of proxy-driven contrastive learning while explicitly addressing pose variance, the dominant challenge in underwater wildlife re-identification.

## 3 METHOD

### 3.1 DATASET: THRESHER SHARK RE-IDENTIFICATION AS A CASE STUDY

Video footage was collected by divers during early morning recreational dives at Kimud Shoal, Malapascua, on January 1, 3, 4, and 5, 2025, a site renowned for frequent thresher shark sightings. Recordings were captured using action cameras such as the GoPro Hero and DJI models. These videos document natural shark behavior in unstructured, open-water environments and serve as the primary data source for the study. Since the footage was captured by 5 divers on the same dates and times, it reflects varied viewpoints and environmental conditions, yet the same set of thresher sharks is likely to appear across multiple videos. This setup increases the likelihood of overlapping individuals across clips, providing a semi-controlled environment that supports re-identification experiments even in the absence of global identity labels. A full bar chart of daily diver contributions is shown in Appendix A.1. Unlike person re-ID benchmarks, these videos lack global identity annotations and are recorded under unconstrained conditions with no fixed cameras.

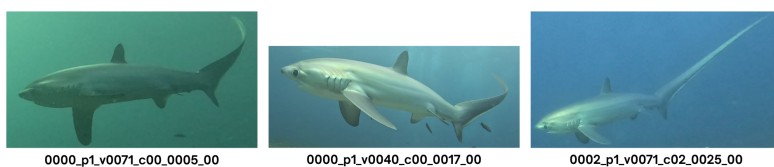

0000_p1_v0071_c00_0005_00    0000_p1_v0040_c00_0017_00    0002_p1_v0071_c02_0025_00

Figure 1: All three images share the same pose label (p1, left-facing), with the first and third showing different sharks from the same dive (v0071, subclips c00 vs. c02), while the second comes from another dive (v0040) and thus has a non-comparable local ID.

Each dive recording is treated as a Video ID (e.g., v0000). To remove ambiguity from overlapping sharks, we manually segment each video into Subclips (c00, c01, ...), such that each subclip contains only a single shark. This guarantees identity purity within subclips and provides reliable tracklets for training and evaluation. Within a video, sharks are annotated with Local IDs (0000, 0001, ...). A Local ID is a video-specific identifier that tracks the same shark consistently across subclips within a single video but cannot be used to match individuals across different videos. For example, in Figure 1, the first and third images originate from the same dive video (v0071) but represent different sharks segmented into distinct subclips (0005 in c00 vs. 0025 in c02). Because they co-occur within the same video, their local IDs are directly comparable. In contrast, the second image comes from a different dive (v0040), so its local ID (0017) is not comparable to those in v0071. Despite these differences, all three images share the same pose label (p1), indicating a left-facing orientation.

Each detection is further annotated with a pose label (p1, p2, p3, p4, p5, p6, p7), corresponding to coarse orientations such as left, right, front-left, front-right, back-left, right, front-right, back-right, and others. These labels serve as proxies for viewpoint, analogous to camera IDs in human re-ID benchmarks, and are crucial for training pose-aware proxies. Finally, frame indices and bounding box IDs provide fine-grained localization within subclips, though in practice each subclip typically

contains a single bounding box per frame. This dataset presents unique challenges: underwater variation due to turbidity, lighting, and distance; non-rigid deformations as sharks swim; and the absence of cross-video global IDs. As such, it serves as a case study for testing re-ID algorithms in ecological video, where assumptions of fixed viewpoints and large labeled datasets do not hold.

### 3.2 POSE-AWARE PROXIES

Our method builds on the Transformer Multi-Grained Framework (TMGF) for unsupervised re-identification. TMGF stabilizes contrastive learning by introducing camera-aware proxies, which exploit the fact that each surveillance camera captures a consistent viewpoint. In our ecological setting, however, no fixed cameras exist. To address this, we introduce pose-aware proxies, replacing camera labels with coarse pose annotations.

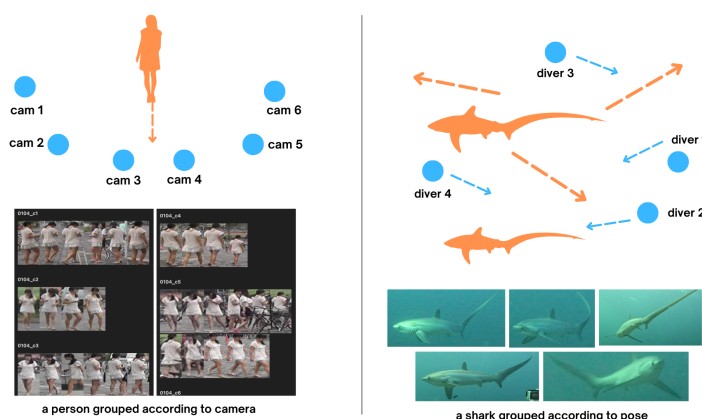

Figure 2: LEFT: In Market-1501, 6 static surveillance cameras naturally provide camera labels as proxies for viewpoint. RIGHT: In our thresher shark dataset, footage is collected opportunistically by divers with handheld action cameras, so we instead annotate each shark by coarse pose labels, which serve as weak proxies for viewpoint.

Concretely, for each image $x_i$, we extract features $f_i = f_\theta(x_i)$ using a ViT-S/16 backbone with multi-grained feature pooling as in TMGF. During each training iteration, clustering assigns pseudo-identities across the dataset. Within each cluster, we further partition samples into subsets according to pose labels. Each subset defines a pose-conditioned proxy $p^{pose}$, which acts as an anchor in contrastive learning. This modification reduces intra-cluster noise by explicitly modeling viewpoint variation.

### 3.3 LOSS FUNCTION

We adopt the contrastive learning objective from TMGF, applied to pose-aware proxies. For an embedding $f_i$ and its corresponding positive proxy $p_i^{pose}$, the loss is:

$$\mathcal{L}_i = -\log \frac{\exp(\text{sim}(f_i, p^{pose}_i)/\tau)}{\sum_j \exp\left(\text{sim}(f_i, p_j^{pose})/\tau\right)}$$

where $\text{sim}(\cdot, \cdot)$ denotes cosine similarity, $\tau$ is a temperature hyperparameter, and the denominator sums over all proxies across clusters and poses. This formulation follows TMGF, but proxies are grouped by pose rather than camera, providing weak supervision for viewpoint without requiring static camera IDs.

The training pipeline proceeds iteratively through three stages: (i) a clustering step, which assigns pseudo-identities across the dataset; (ii) a proxy construction step, which partitions each cluster according to pose labels to form pose-aware proxies; and (iii) a contrastive update step, which optimizes embeddings using the proposed loss function with offline-online association updates. This iterative design preserves the strengths of proxy-based learning while extending its applicability to unconstrained ecological footage.

### 3.4 EVALUATION METRICS

Our method replaces fixed camera IDs with pose-aware proxies that partition pseudo-identities by coarse orientation, aiming to (i) stabilize embeddings within encounters, (ii) harden negative discrimination among co-occurring sharks that share viewpoint/context, and (iii) link the same shark across viewpoint changes within a dive, without cross-video global IDs. The evaluation matches these hypotheses: Within-Track Consistency (WTC) measures temporal stability in identity-pure tracklets, which is a necessary condition for any proxy to be useful. Mutual-Exclusion Error (MEError@k) targets the hardest negatives available, which are other sharks from the same dive; thus, tests whether pose-conditioned supervision reduces confusions precisely where camera-aware proxies help in person Re-ID. Finally, Co-Occurrence Recall (CoR@k) quantifies within-dive linking across subclips; its cross-pose variant isolates viewpoint robustness, the central promise of pose-aware proxies. We therefore read improvements in WTC, MEError, and CoR as evidence that pose-conditioned guidance is functioning as intended under the dataset's partial-label regime.

#### 3.4.1 WITHIN-TRACK CONSISTENCY (WTC)

Each subclip contains only one shark. Let a subclip tracklet be $T = \{f_1, f_2, \ldots, f_m\}$, where $f_i$ are embeddings of frames from the same shark. We measure stability as the mean squared deviation from the tracklet centroid:

$$\text{WTC}(T) = \frac{1}{m} \sum_{i=1}^{m} \|f_i - \bar{f}_T\|_2^2, \quad \bar{f}T = \frac{1}{m} \sum i = 1^m f_i$$

The dataset-level score is the mean over all subclips. Lower values indicate temporally smoother embeddings.

#### 3.4.2 MUTUAL-EXCLUSION ERROR (MEERROR@K)

For a query embedding $f_i$ from shark $s_i$ in video $v$, we define the set of negatives $N_i$ as embeddings of other sharks in the same video (different Local IDs). Let $\mathcal{R}_k(f_i)$ be the top-k retrieved embeddings. The error rate is the fraction of negatives retrieved:

$$\text{MEError@k} = \frac{1}{|Q|} \sum_{f_i \in Q} \frac{1}{k} |\mathcal{R}_k(f_i) \cap N_i|$$

This penalizes cases where different sharks filmed in the same dive are incorrectly retrieved as the same. Lower values are better.

#### 3.4.3 CO-OCCURRENCE RECALL (COR@K)

For a query embedding $f_i$ of shark s in video v, we define the set of positives $P_i$ as embeddings of the same shark (same Local ID) that appear in other subclips of the same video. Retrieval quality is measured as the fraction of these positives that appear in the top-k:

$$\text{CoR@k} = \frac{1}{|Q|} \sum_{f_i \in Q} \frac{1}{|P_i|} |\mathcal{R}_k(f_i) \cap P_i|$$

Higher values indicate that the model successfully links the same shark across different subclips, even with viewpoint and appearance changes.

Pose-aware proxies operationalize viewpoint structure during training by partitioning pseudo-identities into orientation-conditioned anchors. Our evaluation mirrors this structure at test time: WTC verifies per-track stability required by any proxy mechanism, MEError@k measures whether pose-conditioned supervision reduces in-context impostors, and CoR@k, especially in the cross-pose split, tests the promised viewpoint robustness without presuming cross-video labels.

## 4 EXPERIMENTS

### 4.1 DATASET

The dataset comprises 164 clips, each corresponding to a contiguous track of frames, with a median length of 18 frames. To mitigate bias arising from clip length, we stratify the dataset into many-frame groups ( $\geq$ median, 86 clips) and fewer-frame groups ($<$ median, 78 clips). This ensures that both long and short tracks are proportionally represented during training and evaluation, rather than

Table 1: Dataset split by clip length. Stratification ensures balanced representation of both long and short tracks. Queries are formed by randomly sampling one frame per gallery clip.

| Category | # Clips | Training (70%) | Gallery (30%) | Query (1 per gallery) |
|---|---|---|---|---|
| Many-frame ($\geq$ 18) | 86 | 60 | 26 | 26 |
| Fewer-frame ($<$ 18) | 78 | 55 | 23 | 23 |
| **Total** | 164 | 115 | 49 | 49 |

allowing long tracks to dominate. Within each stratum, 70% of clips are randomly assigned to the training set, while the remaining 30% are allocated to the gallery set. From each clip in the gallery, we randomly sample one frame as the query. This design forces the model to generalize across frames within the same track, rather than overfitting to clip-specific redundancies, and it mirrors the retrieval setting where a single observation must be matched against a reference set.

## 4.2 SETUP

We evaluate on the Thresher dataset, curated from diver-captured videos at Malapascua. The dataset comprises 164 clips grouped into local track identities, stratified by clip size (median = 18 frames). We use 70% of tracks for training and allocate the remaining 30% to gallery sets, with a single randomly sampled frame per clip serving as the query set. All images are resized to 128×384 to preserve the elongated, fusiform body shape of thresher sharks, which predominantly swim in a horizontal orientation, and normalized using dataset-specific mean (0.2495, 0.5476, 0.5399) and standard deviation (0.1439, 0.1680, 0.1546). Our backbone is the Transformer-based Multi-Granularity Framework (TMGF) Li et al. (2022) built on ViT-S/16 (L=12, D=384), initialized from LUPerson pretraining. We retain the 5-branch part granularity and patch stride (16×16, yielding 8×24 patches), but replace camera-aware embeddings with pose-aware proxies, defined over seven coarse orientations. Unless otherwise stated, pose supervision strength is set to $\lambda c = 3$. Training follows SGD with momentum (0.9), weight decay ($5 \times 10^{-4}$), and base learning rate ($3.5 \times 10^{-4}$, scheduled with a 10-epoch warmup and decays over 50 total epochs. We use batch size 32, 8 workers, and enable FP16 mixed precision. All experiments run on a single NVIDIA A100 GPU. For unsupervised identity discovery, we adopt DBSCAN with eps = 0.5 and min_samples = 4, coupled with a memory bank (momentum = 0.2) and proxy temperature = 0.07. Sampling is proxy-balanced with 4 instances per identity proxy to stabilize training. Evaluation follows a query-gallery protocol, where each query frame is retrieved against the gallery. We report both conventional metrics (mAP, CMC) and weakly supervised measures tailored to this dataset: within-track consistency (WTC), co-occurrence recall (CoR@k), and mutual-exclusion error (MEError@k). A complete tabular summary of the experiment setup is provided in Appendix A.3.

## 4.3 RESULTS

Table 2: Evaluation results on the Thresher Shark dataset.

| Metric | Random | ImageNet | Base | Pose (TP6) |
|---|---|---|---|---|
| Within-track variance | 0.9264 | 0.1571 | 0.2625 | 0.2009 |
| Mutual-exclusion error @1 | 0.0117 | 0.0117 | 0.0036 | 0.0029 |
| Mutual-exclusion error @5 | 0.0097 | 0.0341 | 0.0213 | 0.0115 |
| Mutual-exclusion error @10 | 0.0098 | 0.0473 | 0.0466 | 0.0309 |
| Co-occurrence recall @1 | 0.0002 | 0.0021 | 0.0088 | 0.0070 |
| Co-occurrence recall @5 | 0.0009 | 0.0098 | 0.0434 | 0.0328 |
| Co-occurrence recall @10 | 0.0021 | 0.0174 | 0.0774 | 0.0713 |

Across 164 identity-pure tracklets, Pose (TP6) improves temporal stability and negative discrimination relative to Base, while ImageNet-pretrained features appear smoother but degrade retrieval, and Random behaves as expected (high variance, almost no positive retrieval). Concretely, TP6 lowers

WTC by 23.5% (0.2625 → 0.2009), and reduces MEError vs. Base by 19.4% @1, 46.0% @5, and 33.7% @10. However, TP6 also yields slightly lower CoR than Base (-20.5% @1, -24.4% @5, -7.9% @10). These results indicate a precision-recall trade-off: pose-aware proxies suppress same-dive impostors more aggressively, but may over-separate cross-pose instances of the same shark, reducing positive coverage at small $k$.

### 4.3.1 TP6 SUBSTANTIALLY REDUCES WITHIN-TRACK VARIANCE

TP6 achieves 0.2009, improving on Base (0.2625) by 23.5%; ImageNet is lower still (0.1571; -40.1% vs. Base), while Random is substantially worse (0.9264; 3.53× Base). The ImageNet result should be interpreted cautiously: very low WTC can reflect generic smoothing (pose/appearance averaging) rather than identity specificity, which is consistent with its weak retrieval below. Methodologically, report macro-averaged WTC over subclips (each tracklet contributes one score) and, if space permits, stratify WTC by subclip length to control for temporal averaging effects. We recommend adding robust uncertainty estimates (e.g., bootstrap confidence intervals over tracklets or a Wilcoxon signed-rank test vs. Base) to confirm the TP6 gain is statistically reliable.

### 4.3.2 TP6 STRONGLY SUPPRESSES IMPOSTOR MATCHES (MEERROR)

Relative to Base, TP6 reduces errors among same-dive negatives by 19.4% @1 (0.0036 → 0.0029), 46.0% @5 (0.0213 → 0.0115), and 33.7% @10 (0.0466 → 0.0309). ImageNet performs worse than Base at $k \geq 5$ (+60% @5, +1.5% @10), aligning with the "smoothing"' hypothesis. Random appears deceptively strong at $k \geq 5$ (0.0097 @5, 0.0098 @10), but this is an artifact of uninformative retrieval (see CoR): it simply does not bring positives into the top-$k$, thereby also avoiding same-dive negatives. To prevent misinterpretation, MEError should always be read together with CoR.

### 4.3.3 POSE-AWARE FEATURES REDUCE CROSS-POSE RECALL (COR)

Base outperforms TP6 at all $k$: @1 0.0088 vs. 0.0070 (-20.5%), @5 0.0434 vs. 0.0328 (-24.4%), @10 0.0774 vs. 0.0713 (-7.9%). ImageNet trails substantially (-76-78% vs. Base across $k$), and Random is near zero. The TP6 drop likely reflects pose specialization: by partitioning clusters into pose-conditioned proxies, embeddings become more pose-discriminative (fewer cross-pose false matches; lower MEError) but slightly less pose-invariant (fewer true cross-pose positives found; lower CoR). The gap narrows by $k = 10$, suggesting that pose-aware features still recover many true positives with modest list depth.

### 4.3.4 TP6 FAVORS PRECISION, BASE FAVORS RECALL

For applications prioritizing precision against impostors within a dive (e.g., expert validation workflows), TP6 is preferable: MEError is sharply reduced with only a small loss in CoR at $k = 10$. If maximizing positive coverage at small $k$ is critical, Base may be competitive. Future studies could quantify this precision-recall trade-off by defining a composite retrieval score computed as one minus the mutual-exclusion error at $k$ multiplied by the co-occurrence recall at $k$, or by plotting mutual-exclusion error versus co-occurrence recall frontiers across retrieval depths of one, five, and ten.

## 4.4 ABLATION

### 4.4.1 EFFECT OF POSE GRANULARITY

Evaluating different pose grouping strategies is essential for understanding how orientation granularity influences the effectiveness of pose-aware proxies. Fine-grained labels may capture subtle viewpoint distinctions but can introduce noise when categories are ambiguous, whereas coarser groupings trade detail for sample balance and robustness. By systematically comparing these variants, we can assess whether performance gains arise from detailed orientation cues or from cleaner, more semantically stable partitions. This analysis ensures that the proposed method is not overly dependent on arbitrary labeling choices and provides insights into the stability of pose supervision under varying levels of granularity. For detailed definitions of each grouping scheme, see Appendix A.2.

Table 3: Pose Granularity results on the Thresher Shark dataset

| Metric | Base | Pose TP2 | Pose TP3 | Pose TP4 | Pose TP6 | Pose TP7 |
|---|---|---|---|---|---|---|
| Within-track variance | 0.2625 | 0.2284 | 0.2725 | 0.2088 | **0.2009** | 0.2629 |
| ME error @1 | 0.0036 | 0.0048 | **0.0026** | 0.0048 | 0.0029 | 0.0036 |
| ME error @5 | 0.0213 | 0.0173 | 0.0197 | 0.0198 | **0.0115** | 0.0151 |
| ME error @10 | 0.0466 | 0.0479 | 0.0517 | 0.0476 | **0.0309** | 0.0356 |
| CoR @1 | 0.0088 | 0.0065 | 0.0074 | 0.0054 | 0.0070 | **0.0079** |
| CoR @5 | **0.0434** | 0.0295 | 0.0359 | 0.0303 | 0.0328 | 0.0361 |
| CoR @10 | **0.0774** | 0.0571 | 0.0668 | 0.0666 | 0.0713 | 0.0645 |

In terms of temporal stability, measured by within-track consistency (WTC), TP6 provides the strongest stabilization among the 164 evaluated tracklets, achieving 0.2009 compared to 0.2625 for Base (-23.47%). TPD also reduces variance (0.2088; -20.46%), while TP2 yields modest improvement (0.2284; -12.99%). By contrast, TP3 slightly worsens WTC relative to Base (0.2725; +3.81%). These results suggest that excluding ambiguous "Others" (P7) systematically improves temporal coherence of embeddings, supporting the hypothesis that P7 introduces label noise that destabilizes tracklet-level features.

For negative discrimination, measured by mutual-exclusion error (MEError@k), TP6 again delivers the strongest improvements over Base, reducing errors by 19.44% at @1 ($0.0036 \rightarrow 0.0029$), 46.01% at @5 ($0.0213 \rightarrow 0.0115$), and 33.69% at @10 ($0.0466 \rightarrow 0.0309$). Pose (unspecified granularity) achieves smaller but consistent gains at $k \geq 5$ (-29.11% at @5, -23.61% at @10), though without improvement at @1. TP3 performs best at @1 (-27.78% vs. Base) but deteriorates by @10 (+10.94%), indicating that coarse lateral merges sharpen immediate nearest-neighbor purity while introducing heterogeneity at deeper ranks. TP2 harms precision at @1 (+33.33%) and produces mixed effects thereafter (-18.78% at @5, +2.79% at @10), highlighting that overly coarse pose supervision sacrifices discriminative structure necessary for impostor rejection.

Positive coverage, measured by co-occurrence recall (CoR@k), reveals the opposite pattern. Base remains strongest overall, particularly at small $k$. TP6 lags behind Base at @1 (-20.45%) and @5 (-24.42%), but the difference narrows at @10 ($0.0774 \rightarrow 0.0713$; -7.88%). Moreover, TP6 outperforms other pose-aware variants at @10 (0.0713 vs. 0.0645 for Pose, 0.0668 for TP3, and 0.0666 for TPD). This reflects the inherent trade-off: by partitioning proxies by orientation, TP6 reduces cross-pose confusion (lower MEError) but at the expense of cross-pose linkage (lower CoR), with the penalty diminishing as the retrieval list deepens.

Taken together, these results indicate that TP6 offers the best precision against same-dive impostors while simultaneously stabilizing temporal embeddings. The associated loss in positive coverage is modest and primarily concentrated at small $k$. Thus, when applications require immediate positive matches, Base remains competitive, but when precision and reviewer workload are critical, TP6 represents the preferable configuration.

The analysis of pose granularity further clarifies how orientation grouping shapes performance. TP6, which excludes the ambiguous P7 "Others" class, consistently outperforms other variants by improving both WTC and MEError with only a small reduction in CoR at $k = 10$. This demonstrates that discarding noisy pose annotations enhances the supervision signal without excessively fragmenting positives. By contrast, TP3, which merges oblique and profile views into coarse left/right buckets while retaining P7, sharpens the top of the ranking (lowest MEError@1) but degrades at deeper ranks (MEError@10 worse than Base, WTC slightly worse). This outcome suggests that mixing oblique and profile orientations inflates intra-proxy variance, undermining retrieval consistency beyond the nearest neighbor. TP2, which collapses all poses into binary left versus right classes, proves too coarse: although it improves WTC, it worsens MEError@1 and yields the steepest decline in CoR. Finally, TPD provides an intermediate trade-off, producing strong WTC and modest MEError gains at @5, but no measurable advantage at @10 and consistently lower CoR than Base. Overall, these results highlight that pose-aware supervision is most effective when ambiguous categories are excluded and orientation granularity is neither too fine nor too coarse.

## 5 CONCLUSION

We empirically evaluate pose-aware proxy supervision for unsupervised thresher-shark re-identification on 164 identity-pure tracklets (1,042 queries). Excluding ambiguous pose annotations (TP6) consistently improves temporal stability and impostor suppression versus a strong clustering baseline. These gains trade off against reduced immediate cross-pose recall, indicating a precision-recall split where pose-homogeneous proxies tighten nearest-neighbour purity but can fragment identity coverage across poses.

Practically, TP6 is the preferred configuration when top-rank precision and temporal consistency matter (e.g., expert validation or conservative population estimates); the Base model may be preferable when maximizing immediate recall at very small k. We caution against using WTC alone for model selection: low WTC can reflect smoothing (ImageNet encoder) rather than identity discrimination, so WTC should be evaluated alongside retrieval metrics (CoR, MEError, mAP/CMC) with uncertainty estimates (bootstrap CIs, paired tests) given our moderate sample size and clip-local evaluation.

Future work should (i) add cross-pose contrastive terms to recover cross-pose positives, (ii) explore hierarchical/soft proxies that balance pose specificity and identity coherence, and (iii) integrate active expert-in-the-loop correction and temporal regularization. These extensions aim to retain TP6's precision gains while improving cross-pose positive coverage and generality.

### 5.0.1 AUTHOR CONTRIBUTIONS

Author 1 was responsible for writing the manuscript, training the model, participating in data collection, and performing data selection and organization. Author 2 provided supervision, guidance, and critical review of the paper.

This work made limited use of large language model (LLM) tools to assist with grammar correction and improving the clarity of writing. All substantive ideas, analyses, and conclusions are those of the authors, who remain fully responsible for the content of the paper.

### 5.0.2 ACKNOWLEDGMENTS

We gratefully acknowledge Evolution Diving, a dive shop based in Malapascua, Philippines, for their valuable assistance in data collection.

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

## A  APPENDIX

### A.1  DAILY DIVERS CONTRIBUTION

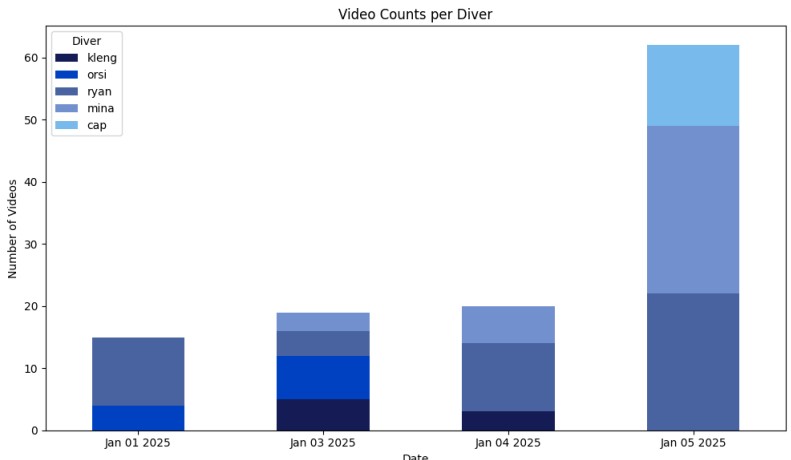

Figure 3: A bar chart of daily diver contributions; diver identifiers are anonymized using aliases.

### A.2  POSE GROUPING VARIANTS

To assess the impact of orientation granularity on pose-aware proxies, we define several grouping strategies derived from the seven original orientation labels: Left (P1), Right (P2), Front-Left (P3), Front-Right (P4), Back-Left (P5), Back-Right (P6), and Others (P7). Below we describe each scheme and its rationale.

Full Granularity (TP7)

- Classes: P1, P2, P3, P4, P5, P6, P7
- This retains the complete taxonomy of poses. Serves as the gold standard and reference baseline, preserving all available pose information.

Excluding Ambiguous Poses (TP6)

- Classes: P1, P2, P3, P4, P5, P6
- This removes the "Others" category (P7), which often contains ambiguous or low-quality examples. It tests whether excluding such noise strengthens the pose signal.

Flank-Separation without Ambiguity (TP4)

- Classes: P1 (Left profile), P2 (Right profile), LeftFlank = P3, P5, RightFlank = P4, P6

- This is the same as A.5 but excludes ambiguous frames (P7). It produces a clean yet semantically structured partitioning, emphasizing the distinctiveness of pure flank views relative to oblique shots.

Lateral-Side Grouping (TP3)

- Classes: Left = P1, P3, P5, Right = P2, P4, P6, Others = P7
- This collapses orientations into semantically interpretable left vs. right flanks while preserving ambiguous cases as "Others." It provides a balance between semantic clarity and sufficient sample size.

Lateral-Side Grouping without Ambiguity (TP2)

- Classes: Left = P1, P3, P5, Right = P2, P4, P6
- This discards ambiguous frames (P7) and reduces orientation to a binary left/right distinction. It tests whether a coarse but strong side-based prior is sufficient for effective supervision.

## A.3 EXPERIMENT SETUP SUMMARY

Table 4: Backbone architecture, training protocol, and clustering configuration.

| **Backbone** | |
| --- | --- |
| Framework | TMGF (ViT-S/16) |
| Transformer depth | $L = 12$ |
| Embedding dimension | $D = 384$ |
| Patch stride | $16 \times 16$ ($8 \times 24$ patches) |
| Granularity branches | 5 |
| Proxy type | Pose-aware (7 orientations) |
| Pose supervision strength | $\lambda_c = 3$ |
| Pretraining | LUPerson |
| **Training** | |
| Optimizer | SGD |
| Learning rate | $3.5 \times 10^{-4}$ |
| Weight decay | $5 \times 10^{-4}$ |
| Momentum | 0.9 |
| Warmup epochs | 10 |
| Total epochs | 50 |
| Batch size | 32 |
| Workers | 8 |
| Mixed precision | FP16 |
| Hardware | NVIDIA A100 |
| **Clustering / Memory Bank** | |
| Clustering algorithm | DBSCAN |
| DBSCAN $\epsilon$ | 0.5 |
| DBSCAN min_samples | 4 |
| Memory bank momentum | 0.2 |
| Proxy temperature | 0.07 |
| Sampling strategy | Proxy-balanced (4 instances/proxy) |

