# OpenReview forum: "Pose-Aware Proxies for Unsupervised Marine Wildlife Re-Identification"
_ICLR.cc/2026/Conference — Submitted to ICLR 2026_

### Official Review · Reviewer_56z6 · 2025-10-30

**Soundness:** 3
**Presentation:** 3
**Contribution:** 2
**Rating:** 2
**Confidence:** 4

**Summary:**

This paper presents a *Pose-Aware Proxy (PAP)* framework for unsupervised animal ReIDentification. The core idea is to replace camera-conditioned proxies with pose-conditioned ones, addressing ecological underwater videos that lack global identity labels. The authors construct a small Thresher Shark dataset and introduce three weakly supervised metrics to evaluate temporal stability, false-match suppression, and within-dive recall. Experiments show that removing ambiguous pose classes (TP6 configuration) improves feature stability and reduces false matches compared with a baseline clustering model.

**Strengths:**

1. The paper tackles a unique and challenging problem: unsupervised ReID for marine animals without global ID labels.
2. The idea of pose-conditioned proxies extends camera-aware proxies to ecological settings, which is an interesting conceptual step.
3. The proposed weakly supervised metrics are internally consistent and address the practical constraints of the dataset.
4. The ablation studies are systematic and their outcomes match the stated intuition.
5. The data come from real underwater footage, making the study grounded and realistic.

**Weaknesses:**

### 1. Evaluation design doesn’t really test ReID ability

Training on unlabeled or partially labeled data is reasonable, but evaluating on a dataset **without any global identity annotations** makes it unclear what is actually being measured.
The three proposed metrics assess feature stability and within-video discrimination, not whether the model can consistently recognize the same individual across videos—which is the essence of ReID.

As a result, the reported results are hard to interpret in terms of real-world identification ability. While the reviewer understands that annotating marine wildlife is difficult, this dataset only targets one species, so collecting at least partial global IDs through more controlled capture or expert annotation seems achievable and would make the evaluation much more convincing.


### 2. Experiments are too narrow

All experiments are conducted only on the authors’ own dataset. There are no comparisons with existing ReID methods, and the model is not tested on any standard animal ReID datasets that already have global identity labels. Many datasets could have been used for cross-dataset evaluation. Even if they involve simpler poses, subsets could be chosen to match this work’s conditions.

Moreover, the point of training with strong pose variations is not to classify poses but to make the model learn fine-grained discriminative cues. It would therefore be valuable to train on this shark dataset and test on other labeled datasets to demonstrate cross-domain generalization.


### 3. Contributions and workload feel limited

The paper essentially offers two main contributions: the pose-conditioned proxy method and the dataset with custom metrics. While using pose as a proxy target is a decent idea, the technical modification is simple and requires much broader experimentation to justify its significance. The dataset is small, around a hundred local IDs and only a few thousand images, and both its **scale** and **annotation precision** are below the field standard.

If expanding the dataset is not feasible, the authors should make better use of what they have—through cross-validation, augmentation, or cross-domain transfer—to show that the findings are not just dataset-specific.


### 4. The “pose as proxy” idea needs stronger evidence

The assumption that pose can reliably serve as a proxy for identity learning is not fully supported. The ablation on pose granularity (TP2–TP7) shows that removing noisy poses helps, but that mainly proves the model is sensitive to label noise. It doesn’t demonstrate that pose-conditioned supervision truly improves identity learning rather than just encoding viewpoint differences.
No theoretical explanation or cross-dataset verification is provided to support this claim, leaving the reliability of the pose-based proxy largely empirical.

**Questions:**

1. How should we interpret WTC, MEError, and CoR as meaningful indicators of ReID performance in the absence of global IDs? Did you check their correlation with conventional ReID metrics (e.g., mAP or CMC) on any labeled data?
2. Have you tried evaluating on existing labeled animal ReID datasets? If not, why, and how do you expect your model to perform there?
3. Could you discuss whether partial global ID labeling would be feasible for your dataset, perhaps through expert tagging or more specific stratigies, to make evaluation more standard?
4. Can you provide additional evidence (theoretical or experimental) to show that pose conditioning genuinely aids identity discrimination rather than overfitting to viewpoint?

---

### Official Review · Reviewer_VYh4 · 2025-10-31

**Soundness:** 2
**Presentation:** 3
**Contribution:** 1
**Rating:** 2
**Confidence:** 4

**Summary:**

The authors propose extending the TMGF framework to work under the more challenging scenario of non-fixed viewpoint cameras. To this end, the authors carry out their study using data from "handheld" video cameras underwater, observing sharks.

The authors claim the following contributions:
1. pose-aware proxies compatible with TMGF pipelines,
2. an ecological evaluation suite aligned to dive-local structure,
3. a realistic underwater case study, and
4. a systematic analysis of pose granularity that informs future unsupervised wildlife reID designs.

No further contributions beyond these are apparent.

**Strengths:**

# Originality

The study does look at a novel, challenging problem

# Quality

The quality of the presentation is good.

# Clarity

The paper falls apart here. Several terms are introduced without explanation of their meanings are and this only becomes apparent much later.
e.g. "treating viewpoint as a structured nuisance and mirroring"

# Significance

The results are significant to the wildlife community, but lack the robustness necessary for broader applicability.

**Weaknesses:**

# Contribution lacks novelty

Of the four contributions, only contribution 1 is relevant to the ICLR community. The others are more applicable in the specific domain.
1. pose-aware proxies compatible with TMGF pipelines,

# Experiments are insufficient

The use of a single dataset to make a claim is somewhat tenuous. Also, the only baseline used is TMGF, which has long since been surpassed in the literature [https://scholar.google.com/scholar?cites=9230028336061960417&as_sdt=2005&sciodt=0,5&hl=en&oi=gsb].
None of these newer methods is considered.
[https://ieeexplore.ieee.org/abstract/document/10382666?casa_token=VyNRk8BVr_EAAAAA:Pj3sM1JtAJ38G-VGxuyKPnpq6x2wR6MkDZv-l4dGs63m5unpbTnGipZOHuWCQ82k_ZLtgpGwqU0]

# Not reproducible

Notably, the paper does not contribute the code or dataset used to the community. This means that the study is not reproducible.

**Questions:**

Why are none of the object tracking frameworks compared against?

---

### Official Review · Reviewer_4La5 · 2025-10-31

**Soundness:** 3
**Presentation:** 3
**Contribution:** 2
**Rating:** 2
**Confidence:** 4

**Summary:**

This paper addresses the challenge of unsupervised marine wildlife re-identification (Re-ID), where conventional fixed camera IDs and global identity labels are limited. The authors propose Pose-Aware Proxies, which replace camera-aware proxies in the Transformer Multi-Grained Framework (TMGF) with coarse pose orientation labels (e.g., left, right, front-left). This modification allows the model to provide weak supervision for viewpoint normalization in unconstrained diver-captured footage.

**Strengths:**

1.	The focus on marine wildlife Re-ID under weak supervision is relevant to ecological monitoring applications.
2.	Pose-aware proxies are a clean and interpretable way to replace camera-aware conditioning without requiring extra sensors or labels.
3.	The WTC, MEError, and CoR metrics are well-motivated and tailored to unlabeled, unstructured ecological videos.

**Weaknesses:**

1.	Dataset scale and generalization: The proposed dataset (164 clips, 18-frame median) is too small to demonstrate generalization. Including more marine species would greatly enhance the dataset’s scalability and ecological relevance.
2.	Limited methodological novelty: The main modification is replacing camera-aware conditioning with pose-aware conditioning which is conceptually straightforward and well-motivated, but lacks deeper algorithmic innovation.
3.	Manual pose annotation: Pose labels are manually assigned. It would be valuable to explore whether existing animal pose estimation frameworks [1][2] could be adapted or lightly fine-tuned on a small subset of annotated frames to build a pose estimation model specific to this dataset, enabling more automated and scalable orientation labeling.
[1] Mathis A, Mamidanna P, Cury K M, et al. DeepLabCut: markerless pose estimation of user-defined body parts with deep learning[J]. Nature neuroscience, 2018, 21(9): 1281-1289.
[2] Cao J, Tang H, Fang H S, et al. Cross-domain adaptation for animal pose estimation[C]//Proceedings of the IEEE/CVF international conference on computer vision. 2019: 9498-9507.

**Questions:**

1.	Have the authors considered using existing pose estimation methods to automatically generate orientation labels, thereby reducing the time cost and potential bias of manual annotation?
2.	Could the authors provide additional visualizations of the dataset and qualitative retrieval results (e.g., top-5 examples) to better illustrate what kinds of similarities the pose-aware model captures?
3.  Could the authors discuss or test the generalization of the proposed pose-aware proxy framework on other marine species or ecological datasets?

---

### Official Review · Reviewer_H1WL · 2025-11-04

**Soundness:** 3
**Presentation:** 3
**Contribution:** 2
**Rating:** 4
**Confidence:** 4

**Summary:**

This paper addresses unsupervised re-identification of thresher sharks in unconstrained underwater videos, where traditional camera-aware proxy methods fail due to handheld, drifting cameras and lack of global identity labels. The authors propose pose-aware proxies—coarse orientation labels (e.g., left, right, front-left)—to provide weak viewpoint supervision within a clustering-based contrastive framework. They curate a structured dataset of 164 identity-pure shark tracklets from Malapascua dive footage and introduce three field-aligned evaluation metrics: Within-Track Consistency (WTC), Co-occurrence Recall (CoR@k), and Mutual-Exclusion Error (MEError@k). Experiments show that the TP6 variant (excluding ambiguous "Others" pose class) improves temporal stability (−23.5% WTC) and reduces impostor matches (MEError down 19.4% @1, 46.0% @5), albeit with a slight drop in CoR at small k that narrows by k=10.

**Strengths:**

* Novel and ecologically grounded problem: Tackles a real-world conservation challenge with practical impact.

* Well-motivated method: Pose-aware proxies elegantly adapt camera-conditioned unsupervised re-ID to ecological settings.

* Thoughtful evaluation: The proposed metrics directly reflect dive-local structure and precision-recall trade-offs relevant to field biologists.

* Rigorous ablation: Systematic analysis of pose granularity (TP2–TP7) shows that excluding ambiguous poses (TP6) yields optimal balance.

**Weaknesses:**

* Limited dataset scale: Only 164 tracklets from a single species/site; generalizability to other marine animals or environments is unverified.

* Coarse pose labels: Manual annotation of 7 orientation classes may not scale and could introduce subjectivity.

* Trade-off not fully resolved: Improved precision comes at the cost of cross-pose recall, highlighting a key limitation the method doesn’t overcome.

* Baseline comparison: Lacks comparison to recent unsupervised animal re-ID methods (e.g., MiewID, ALFRE-ID).

* Missing references: some important wildlife re-ID benchmark are missing in this paper, for instance
> Li, S., Li, J., Tang, H., Qian, R., & Lin, W. (2019). ATRW: a benchmark for Amur tiger re-identification in the wild. arXiv preprint arXiv:1906.05586.

**Questions:**

* How would pose-aware proxies perform on species without clear bilateral symmetry or with more complex pose dynamics?

* Could pose estimation be automated (e.g., via keypoint detection) to eliminate manual labeling?

* What is the sensitivity to the number of pose classes? Is there an optimal granularity beyond TP6?

* How does performance vary across different underwater conditions (e.g., turbidity, lighting)?

---

### Meta-Review · Area_Chair_oChx · 2026-01-08

**Summary:**

While reviewers broadly agreed that the problem is novel, ecologically important, and thoughtfully evaluated within its constraints, the consensus concern was that the technical contribution and experimental evidence fall short of ICLR’s expectations for generality, novelty, and rigor. In particular, the presented work is too narrow and incremental to justify acceptance without broader validation, stronger baselines, and clearer evidence to advance unsupervised Re-ID beyond this specific case study.

**Reviewer Concerns:**

There is no rebuttal.

**Reviewer Scores:**

There is no rebuttal.

---

### Decision · Program_Chairs · 2026-01-26

Reject